

# Quality assessment of the commercially available alcohol-based hand sanitizers with femtosecond thermal lens spectroscopy

Subhajit Chakraborty[1,2], Ashwini Kumar Rawat[3], Amit Kumar Mishra[3] and Debabrata Goswami[1,3]

[1] Centre for Lasers & Photonics, Indian Institute of Technology Kanpur, Uttar Pradesh, India
[2] School of Chemistry, The University of Melbourne, Parkville, Victoria, Australia
[3] Chemistry, Indian Institute of Technology, Kanpur, Uttar Pradesh, India

## ABSTRACT

Using femtosecond-pulse-induced thermal lens spectroscopy (FTLS), we report a novel method for the quality measurements of alcohol-based hand sanitizers (ABHS). To sustain its effectiveness, the ABHS must contain the recommended concentration of alcohol content. We diluted the hand sanitizer with water to reduce the quantity of alcohol in the mixture and then performed thermal measurements on it. We performed both dual-beam Z-scan and time-resolved TL measurements to identify the alcoholic content in the ABHS. The thermal lens (TL) signal of the solvent is capable of detecting any relative change in the alcohol content in the mixture. Our technique, therefore, emerges as a sensitive tool for quality testing of alcohol-based hand sanitizers.

## INTRODUCTION

Hand sanitizers have become a part of our daily life post COVID-19 pandemic. There are primarily two types of hand sanitizers, alcohol free hand sanitizers and alcohol-based hand sanitizers (AHBS). The alcohol-free sanitizer uses chemicals having antiseptic qualities to achieve its antimicrobial effects. Depending on their chemical functional groups, these compounds operate and function in various ways (*Bloomfield & Arthur, 1994*; *McDonnell & Russell, 1999*). ABHS may include one or more types of alcohol, together with additional excipients and humectants. Without the need for water or drying with towels, ABHS can effectively and rapidly eliminate a broad spectrum of microorganisms (*Lee et al., 2020*). ABHS formulations on the market consist of low viscosity liquids, gels, foams, dispensers, and wipes (*Matatiele et al., 2022*). The efficacy of ABHS is reliant on the interaction of multiple parameters, including alcohol type and amount, formulation, other components, manufacturing method, and correct usage technique (*Abuga & Nyamweya, 2021*).

Hand sanitizers are typically composed of both active and inactive ingredients. Active ingredients are the compounds in the hand sanitizer (ABHS) that are responsible for killing germs and bacteria on the skin. The most common active ingredient in hand

Corresponding author
Debabrata Goswami,
dgoswami@iitk.ac.in

sanitizers is alcohol, typically at a concentration of 60–95%. Inactive ingredients, also called excipients, are the compounds in the hand sanitizer that are not responsible for killing germs and bacteria, but they help to form the final product, enhance the stability of the active ingredients, and improve the texture and smell of the product. Some examples of inactive ingredients found in hand sanitizers include water, glycerin, propylene glycol, fragrances, and colorants (*Golin, Choi & Ghahary, 2020*; *Marumure et al., 2022*).

Inactive ingredients such as glycerin and propylene glycol are often used in hand sanitizers to improve the texture and consistency of the product, making it easier to apply and spread over the skin. Some inactive ingredients, such as antioxidants and pH buffers, are used to help stabilize the active ingredients and prevent them from breaking down or losing effectiveness over time. Fragrances and colorants are used to improve the appearance and smell of the product, making it more pleasant to use. Some inactive ingredients are used as preservatives to improve the product's shelf life and prevent it from being contaminated by bacteria or other microorganisms (*Jing et al., 2020*; *Park et al., 2010*).

Measuring the alcoholic content of hand sanitizers is crucial for ensuring their effectiveness in preventing the spread of germs and bacteria. Studies have found that solutions with 60–95% alcohol are most effective in killing germs on the hands. The 60% threshold is considered a minimum effective concentration, above which the alcohol becomes more effective in killing germs. It denatures the proteins and dissolves the lipid membranes of microorganisms, such as bacteria and viruses. When applied to the surface of the skin, alcohol disrupts the cell membrane, causing the contents of the cells to leak out, which leads to the death of the microorganism (*Jing et al., 2020*; *Prajapati, Desai & Chandarana, 2022*; *Han et al., 2022*; *Matatiele et al., 2022*; *Marumure et al., 2022*; *Filipe et al., 2021*; *Chojnacki et al., 2021*).

The alcohol content of ABHS products ranges from 60 to 95% (v/v). The US Centers for Disease Control and Prevention (CDC) recommends a concentration of 60–95% ethanol (EtOH) or 2-propanol (IPA) mixed with distilled water for alcohol-based hand sanitizers (*Matatiele et al., 2022*; *Boyce, 2002*). The World Health Organization (WHO) has advised using alcohol-based hand sanitizers in the absence of water to prevent the spread of the coronavirus since the SARS-CoV-2, COVID-19 outbreak (*World Health Organization, 2020*).

Because of this recommendation, there is much demand for ABHS, and because of the heavy demand for these sanitizers, it is often observed that the hand sanitizers' quality is compromised by reducing the alcoholic quantity in the solution (*Yusuf, 2021*). The effectiveness of an ABHS depends on its alcohol concentration; therefore, quality control is crucial to maintain product integrity and to ensure that consumers are purchasing and using products that have virucidal activity against COVID-19. It is therefore critical to develop simple and quick measurement techniques for detecting alcohols in ABHS as well as determining the alcohol content in commercial ABHS to verify that the user is purchasing an effective product.

A few previous attempts have been made for the requirement of quality control (*Gupta, Rodriguez & Yilmaz, 2021*; *Abrigo et al., 2022b*; *Abrigo et al., 2022a*). To check for impurity chemicals, hand sanitizers were examined using a qualitative nuclear magnetic resonance

(NMR) method (*Bedner et al., 2021*). The gas chromatographic method was utilized in order to measure the amount of alcohol present in both commercially available and homemade ABHS that were liquid and gel-based, respectively (*Yusuf, 2021*; *Abuga, Nyamweya & King'ondu, 2021*). The quality of the ABHS was also evaluated using NIR spectral data analysis (*Pasquini et al., 2020*).

However, most of these analytical measurements require rigorous experimental treatment. The NMR method demands an expensive instrumentation capability. Other measurement techniques require external calibration and a sensitivity issue exists for low level impurities (*Bedner et al., 2021*). We therefore propose a simple yet powerful approach to examine the alcoholic content in a sanitizer using thermal lens (TL) spectroscopic technique (*Shen, Lowe & Snook, 1992*). The recent work aims to fulfill the requirement of a simple yet powerful tool for quality measurements of these sanitizers.

Thermal lens spectroscopy is a widely used tool for understanding thermo-optical properties (*Chakraborty et al., 2021*; *Singhal & Goswami, 2020*; *Jiménez-Pérez et al., 2016a*; *Jiménez-Pérez et al., 2016b*; *Oliveira et al., 2020*), Nonlinear optical properties (*Ventura et al., 2021*; *Rodriguez & Chiesa, 2011*), trace analysis (*Shokoufi & Hamdamali, 2010*), photochemical reactions (*Herculano et al., 2011*; *Astrath et al., 2009*), molecular isomerism (*Kumar, Dinda & Goswami, 2014*), distinguishing isotopes (*Bhattacharyya, Kumar & Goswami, 2014*), *etc*. This spectroscopic tool has already been used to learn about the structural information of molecules (*Mikheev et al., 2016*; *Kumar & Goswami, 2014*) and intermolecular interactions (*Chakraborty, Rawat & Goswami, 2019*; *Bhattacharyya, Kumar & Goswami, 2011*).

Thermal lens spectroscopy is a highly sensitive non-destructive spectroscopic technique that can take advantage of localized photothermal heating in liquids. Photothermal heating creates a temperature gradient and a refractive index gradient. The sample then starts to act like a thermally-induced lens. The collimated probe beam encounters spatially modified refractive index inside the sample resulting into modulating the wavefront. In a time-resolved thermal lens experiment, we quantify the probe beam's relative change due to the pump beam. The measurement continues until the sample reaches a steady state or a thermal equilibrium (*Gordon et al., 1965*; *Hempelmann & Dorfmueller, 1991*; *John et al., 2019*).

Femtosecond thermal lens spectroscopy (FTLS) is a specific application of thermal lensing that uses ultrashort laser pulses, typically on the order of femtoseconds, to rapidly heat a sample and create the thermal lens effect. The high repetition rate of these pulses allows for continuous measurement of the sample and makes FTLS a highly sensitive and non-destructive technique for studying photothermal signatures in liquids and mixtures. Since high repetition rate (HRR) femtosecond lasers require around ten times smaller average powers than the CW lasers, there is no excessive heating of the bulk media using this approach within appropriate experimental timeframes. The strong signal-to-noise ratio (SNR) of low power femtosecond HRR lasers makes them ideal for studying TL. Due to the small amount of energy delivered by each femtosecond laser pulse to the sample, the high SNR is feasible. In heavily absorbing samples, localized heat accumulation owing to incident femtosecond pulses causes a superheating state, which also alters the

sample's nonlinear properties leading to a heat dissipation through both convective and convective heat transfer processes (*Mian, McGee & Melikechi, 2002*; *Falconieri & Salvetti, 1999*; *Nóvoa-López et al., 2014*; *Stavrou et al., 2020*; *Kumar Rawat et al., 2022*).

## EXPERIMENTAL DETAILS

The TL measurements were performed by a mode mismatched pump-probe spectroscopic arrangement. A dual output (1560 nm & 780 nm) Er-doped fiber laser (Femtolite; IMRA Inc., Ann Arbor, MI, USA) was employed as the light source. The 1560 nm beam is used as the pump beam, whereas the 780 nm line was used as the probe source. The pump beam is focused using a convex lens for photothermal heating, and the probe beam is kept collimated over the entire path. The schematic diagram of the experimental setup is given in Fig. 1. For the dual beam z-scan experiment, we placed the sample on a motorized translational stage (Newport ESP 300), and then the stage was moved along the pump beam direction. The transmittance of the probe beam as a function of the sample beam was then recorded in a large area photodetector (PDA 100A-EC; Thorlabs, Newton, NJ, USA). A mechanical shutter (SR-475; Stanford Research, Stanford, CA, USA) is placed on the pump beam arm, which allows the pump beam to be incident for a period of 5 s and then blocks its incidence for the next 5 s. Now for the time resolved TL measurements, the sample is positioned at the focus of the pump beam, and then we record the transient behavior of the transmitted probe beam. When the probe beam is allowed to fall on the sample, the transmittance of the beam through the aperture after passing the sample is measured using an amplified silicon photodiode (PDA 100A-EC; Thorlabs, Newton, NJ, USA). Only a fraction of the beam falling on the aperture, which has a 40% opening, reaches the detector. By maintaining the proper distance between the sample and the aperture, the far-field diffraction limits (Fresnel number approximations) are met. Before the detector, a cut-off filter is positioned to block the 1560 nm pump beam, enabling only the 780 nm light to enter the detector. The amplified silicon photodiode is connected to an oscilloscope (LeCroy Wave Runner 64xi, 600 MHz), which is then controlled by a customized LabVIEW software. The lab instruments and data collection are automated using the LabVIEW application.

A UV-Vis-NIR spectrometer (PerkinElmer Lambda 900; PerkinElmer, Waltham, MA, USA) has been used to detect the absorption spectra for each sample. A quartz cuvette with a one mm path length is used to collect the samples, and a resolution of 1 nm is set to acquire the absorption spectra.

### Materials & sample preparation

In our experiment, we used three different alcohol-based sanitizers (S1, S2, and S3), which were further diluted with deionized water before performing the experiments. S1, S2, and S3 are commercially available alcohol based hand sanitizers that were purchased directly from the market. S1 is an IPA based sanitizer whereas the active constituent of S2 is ethyl alcohol (EtOH), and the sanitizer S3 is a mix of both IPA & EtOH.

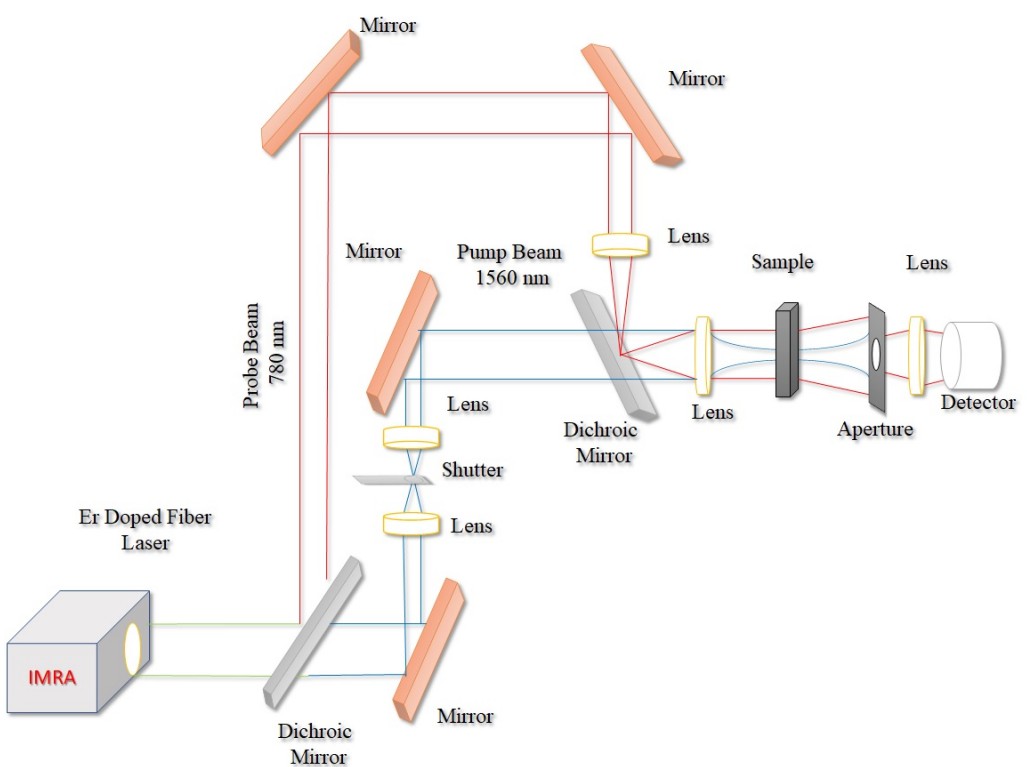

**Figure 1  Schematic experimental setup.** Schematic diagram of the experimental setup for dual beam Z-scan and time resolved measurements.

The sanitizer S1 (SWASA; E-Spin NanoTech Pvt. Ltd., Kanpur, India) is made of isopropyl alcohol (IPA, 70% V/V), purified water, and brilliant blue FCF as a coloring agent.

The sanitizer S2 (SWACHH; Austro Labs Ltd., Delhi, India) has the ingredients neem (Azadirachta Indica), rhizome aloe vera (aloe barbadensis), and ethyl alcohol 85% V/V. The sanitizer S3 (Savlon; ITC Limited, Kolkata, India) contains 70% V/V ethyl alcohol (EtOH) denatured with 3.5% V/V isopropyl alcohol (IPA). The other ingredients include propylene glycol, fragrance, glycerin, disodium EDTA, sodium hydroxide, *etc.*

These three sanitizers and the other two pure solvents (EtOH & IPA) are mixed with DI water at desired volumetric proportions. EtOH, IPA & DI water are of HPLC grade (>99% Purity) and purchased from Sigma Aldrich-Merck, Inc, St. Louis, MO, USA. These sanitizer-water and alcohol-water mixtures were further sonicated at room temperature for 15 min before spectroscopic investigations.

## Mathematical background and theoretical foundations of thermal lens models

The TL signal for dual beam Z-Scan is written in the form of

$$S(z, t_\infty) = \frac{T(z, t_\infty) - T_0}{T_0}. \tag{1}$$

Here $T(z, t_\infty)$ isthe transmittance of the probe beam through the aperture in the presence of the pump beam. $T_0$ isalso the same probe beam transmittance but when the pump beam is absent (*Kumar, Dinda & Goswami, 2014*).

We recently demonstrated that the most widely used model proposed by *Shen, Lowe & Snook (1992)* is incapable of explaining the unusual thermal lens signatures of alcohols with strong absorption at the pump wavelength, causing a greater heat load inside the sample (*Singhal & Goswami, 2020*; *Shen, Lowe & Snook, 1992*). Later, a new model was developed to account for the thermal lens behavior of highly absorbing samples by incorporating both conductive and convective heat transfer modes (*Kumar, Khan & Goswami, 2014*).

The time-resolved TL signal is defined as follows (*Singhal & Goswami, 2019*)

$$S(t) = \frac{I(t)}{I(0)}. \tag{2}$$

Where, $I(t)$ isthe intensity of the probe beam transmitted through an aperture when the pump beam is turned on, whereas $I(0)$ is the intensity of the probe beam transmitted through the aperture when the pump beam is turned off. Now, we can express the steady-state thermal lens signal as

$$S(\infty) = \frac{I(\infty) - I(0)}{I(0)}. \tag{3}$$

Here $I(\infty)$ is considered to be the TL signal after a sufficiently long period of time has passed with no change in the TL signal. The TL signal is additionally written as.

$$TL(t) = 1 - S(t). \tag{4}$$

The following Modified Shen model equation describes the time resolved thermal lens.

$$\frac{I(t)}{I(0)} = \left[ 1 - \frac{\theta_1 + \theta_2}{2} tan^{-1} \left\{ \frac{2mv}{\left((1+2m)^2 + v^2\right)\frac{t_c}{2t} + 1 + 2m + v^2} \right\} \right]^2$$
$$+ F \left[ \frac{\theta_1 + \theta_2}{4} \ln \left( \frac{[1 + 2m/(1 + 2t/t_c)]^2 + v^2}{(1+2m)^2 + v^2} \right) \right]^2. \tag{5}$$

The quantities $\theta_1, \theta_2$ and $t_c$ contain all of the sample's properties that affect the strength of the thermal lens. The geometrical parameters are denoted by m and $v$. which is defined as, $m = \frac{\omega_p^2}{\omega_e^2}$ and $v = \frac{z_1}{z_c} + \frac{z_c}{z_2}\left[ 1 + \left(\frac{z_1}{z_c}\right)^2 \right]$.

$\theta_1 = -Al\left(\frac{dn}{dT}\right)/\lambda_p\kappa$, and $\theta_2 = \left[(\alpha(P-A))l\left(\frac{dn}{dT}\right)/\lambda_p h\right]\exp(-t_d/t) = \theta_{\text{conv}}\exp(-t_d/t)$, $\frac{dn}{dt}$ is the thermo-optic coefficient of the sample; the wavelength of the probe beam is denoted by $\lambda_p$, the sample path length is denoted by $l$, and the absorption coefficient is denoted by A. $\omega_e$ denotes the radius of the pump beam, and $\omega_p$ denotes the radius of the probe beam at the sample position; $P_e$ denotes the power of the pump beam; $t_c = \frac{\omega_e^2}{4D}$, denotes the characteristics time constant, and D denotes the thermal diffusivity.

## EXPERIMENTAL RESULTS

The UV-Vis-NIR absorption spectroscopy (Fig. 2) gives us an idea of the absorption coefficient of the samples at the pump wavelength. The optical absorption of the pump wavelength is very critical to TL measurements as it is directly correlated to the photothermal heating inside the samples. Initially, we took three sanitizers in their pure state and then added DI water in order to dilute the alcohol quantity in the newly prepared mixture. After dilution, the volumetric ratio of the sanitizer and water in the immediate next mixture was set to 90:10. Further, we added water to set the ratio to 80:20, and then again, the amount of water in the mixture was increased in intermediate steps of 10%. All three sanitizers and also EtOH & IPA exhibit significant absorption at 1560 nm (pump wavelength) (*Kumar Rawat et al., 2022*). Among these, water has the highest absorption value, and with the increasing quantity of water in the mixtures, the absorption at the pump wavelength enhances. This results in an increment of thermal heat loading in the system during the TL measurements.

Figure 3A depicts the dual beam Z-scan traces for the pure solvents (viz. EtOH, IPA & water). Among these solvents, IPA & EtOH exhibits a very strong TL signature; however, water has a very low TL signal because of its very high thermal conductivity (*Simpson et al., 2018*; *Korson, Drost-Hansen & Millero, 1969*). In the case of EtOH & IPA, when the sample is moved towards the focal plane, the thermal load inside the system increases accordingly, resulting in an enhanced lensing effect. Therefore, the TL signal is found to be increasing as the sample moves closer to the focal region. Near the focal region, the beam diameter of the pump beam is extremely small, so the intensity of the pump beam is very large, and so is the heat load. As a consequence of the immense heat load, the convective mode of heat transfer comes into the picture as a part of the heat dissipation process. Now because of these convective processes, we observe a slow rise in the TL signal near the focal region. The TL signal exhibits a "W" type signature because of the convective heat transfer mechanisms. Figure 3B represents the dual beam Z-scan traces of the three sanitizers. As the main constituent of these sanitizers is either IPA or EtOH, all of them are highly sensitive to TL effects and exhibit a very strong TL signal. We can also observe the "W" type TL signature for the sanitizer S2, which is evident as the main constituent of S2 is EtOH.

Time resolved TL measurements for the pure solvents are shown in Fig. 3C. In the presence of the pump beam, EtOH & IPA exhibit a strong TL signal. The TL signal for these two solvents reaches a steady state after passing through the inflection point. The rise of the TL signal from the inflection point to the steady state is a result of the convective heat transfer processes arising in the system. Water has a comparatively weaker TL response than the other two, which could be attributed to its exalted thermal conductivity. Fig. 3D represents the time resolved TL signal for all three sanitizers under investigation. As the main constituents of these sanitizers respond positively to the thermal tensing effects, the sanitizers are also highly sensitive to TL effects and exhibit a very strong TL signal during time resolved measurements.

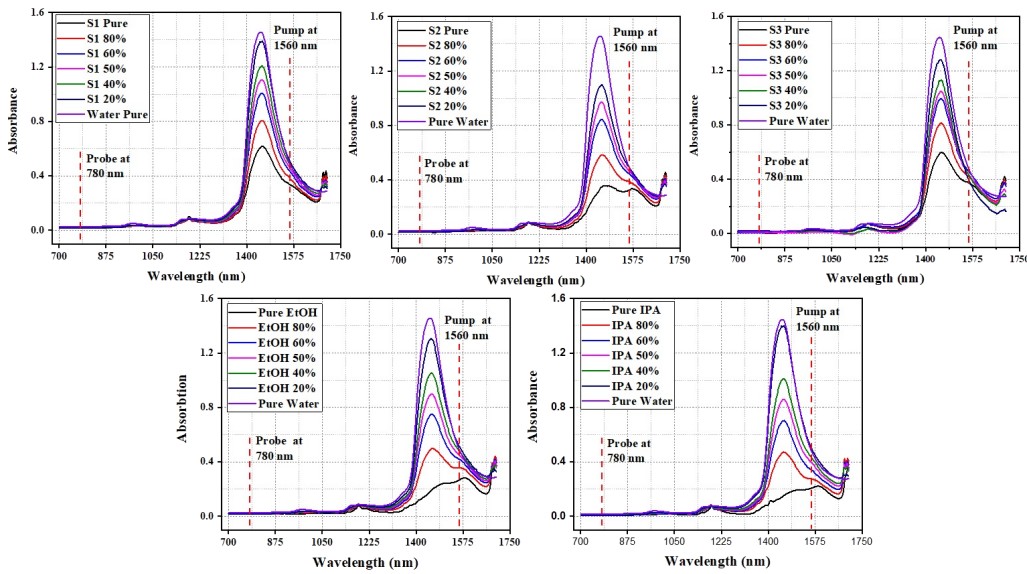

**Figure 2** **Spectroscopic characterization of the samples.** UV-Vis-NIR absorption spectra of the samples.

Figures 4A, 4B, and 4C represent the dual beam Z-scan traces of the sanitizer S1, S2, and S3, respectively. In all three figures, we observe that the pure sanitizers and their diluted mixtures are very much sensitive to the TL measurements. During the dual beam Z-scan the maximum TL response is observed near the focal plane. The amplitude of the TL signal for the pure solvents is found to be maximum for all three cases. Now upon addition of water to the mixture reduces the volume fraction of the sanitizer component in the mixture. From the UV-Vis NIR absorption spectrum (Fig. 1), we notice that the absorbance value of the mixture at the pump wavelength is increased with the increasing concentration of water in the mixture. As the absorption increases, we expect the thermal heat load deposited to the system is also be increasing in a similar fashion. The enhanced heat load should contribute to a greater TL signal amplitude, but on the contrary, we observe that the TL signal decreases with increasing concentration of water.

This could be attributed to the very high thermal conductivity of water. Water systems have a very strong hydrogen-bonded network. Consequently, the deposited heat could be transported very effectively in the system, and hence the TL signal decreases although the heat load increases in the system. The highly conductive hydrogen-bonded network of the water molecules takes care of the transportation of the accumulated energy.

Earlier in Fig. 3, we have observed that the main constituents of the sanitizers, *i.e.,* IPA & EtOH, exhibit a convective property, and so are the pure sanitizers. In Fig. 4, also we notice a W-type signature in the z scan traces of the samples for pure sanitizers. However, this feature diminishes upon mixing water in the system. This indicates that the convective effects slowly die out, and conduction is established as a dominating mode of heat transportation in the system. When the sanitizer concentration falls below 60% (v/v), the TL signal amplitude is found to fall very sharply, and the TL response of the system is diminished to much extent when the water content is significantly high in the mixture.

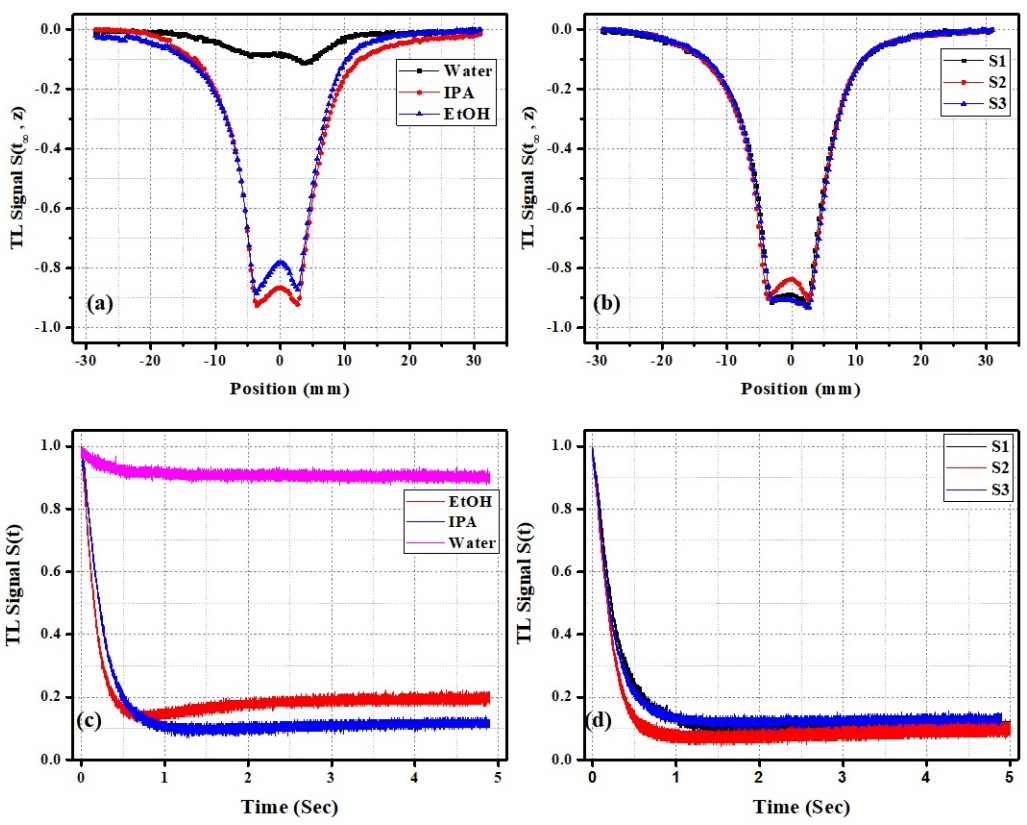

**Figure 3** **(A–D) Measurements of pure solvents.** Dual beam Z-scan and time resolved TL measurements of pure solvents.

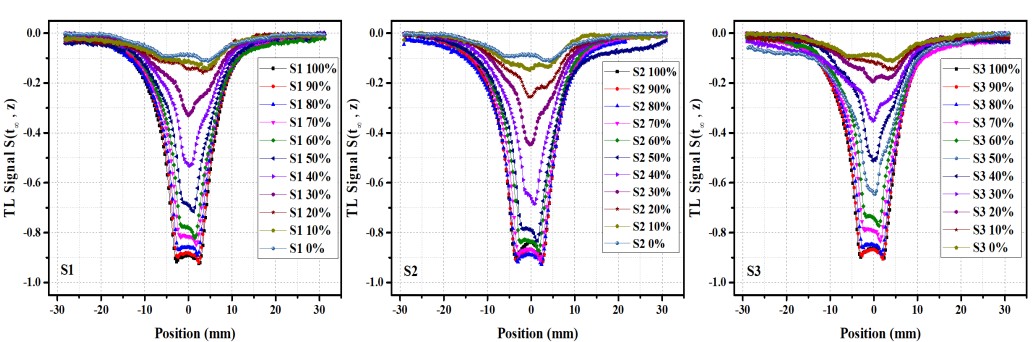

**Figure 4** **Thermal lens measurements of sanitizers.** Dual beam Z-scan and time resolved TL measurements of sanitizers and their mixtures with water.

Figs. 5A, 5B and 5C represent the time resolved TL signal of the sanitizers and their diluted mixtures with water. Pure sanitizers exhibit the maximum steady state TL signal. The steady state TL signal is found to be decreasing when water is added to the system. Very similar to the dual beam z scan traces, the sharp fall in the steady state TL signal is observed

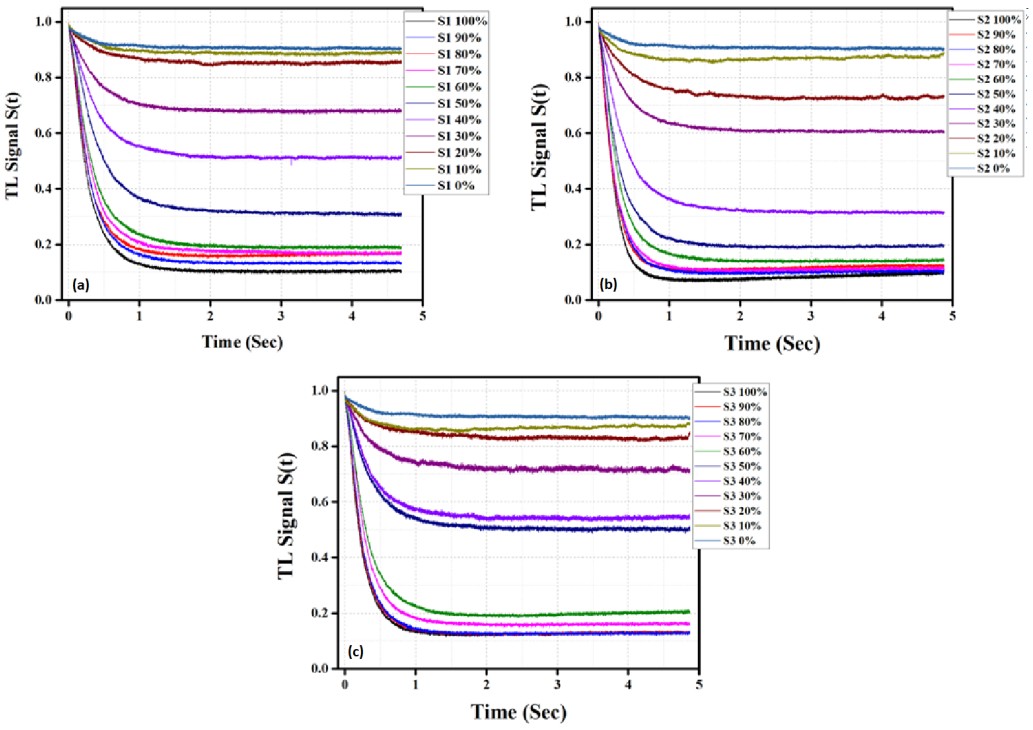

**Figure 5** **Time resolved TL data.** Time resolved TL signal of (A) sanitizers and their various mixtures with water (B) and (C).

when the sanitizer concentration in the system falls below 60% (v/v). At very low sanitizer concentrations, the TL response is also negligible in the time resolved measurements.

Till now, we have understood the TL signatures of the pure sanitizers and their main constituents. We have also explored the TL response of the sanitizers when they were diluted with water. However, to understand the TL response in more detail, it is necessary to comprehend the dilution effects on the main constituents of the sanitizers. We further prepared mixtures by diluting EtOH and IPA with water and performed TL measurements with these samples.

Figure 6A depicts the dual beam z scan traces for the IPA-water mixtures. We observe a "w" type feature in the z scan trace of pure IPA, similar to Fig. 3, which is indicative of the presence of convective characteristics in the system. Surprisingly the pure IPA does not exhibit the maximum TL signal amplitude. When water is added to the mixture, we observe the TL signal amplitude increases up to some extent and then decrease upon further addition of water. The convective feature also diminishes slowly when water is added to the system establishing the conductive mode into a dominating mode of heat transfer when the water proportions are significantly large in the system.

Figure 6B represents the z scan traces for EtOH-water mixtures. EtOH also exhibits a very strong convective heat transfer feature in its TL signature. However, upon adding water in the system, these convective effects are diluted, similar to the earlier ones. Also, in the TL response of the EtOH-Water mixture, the amplitude of the TL signal is not maximum

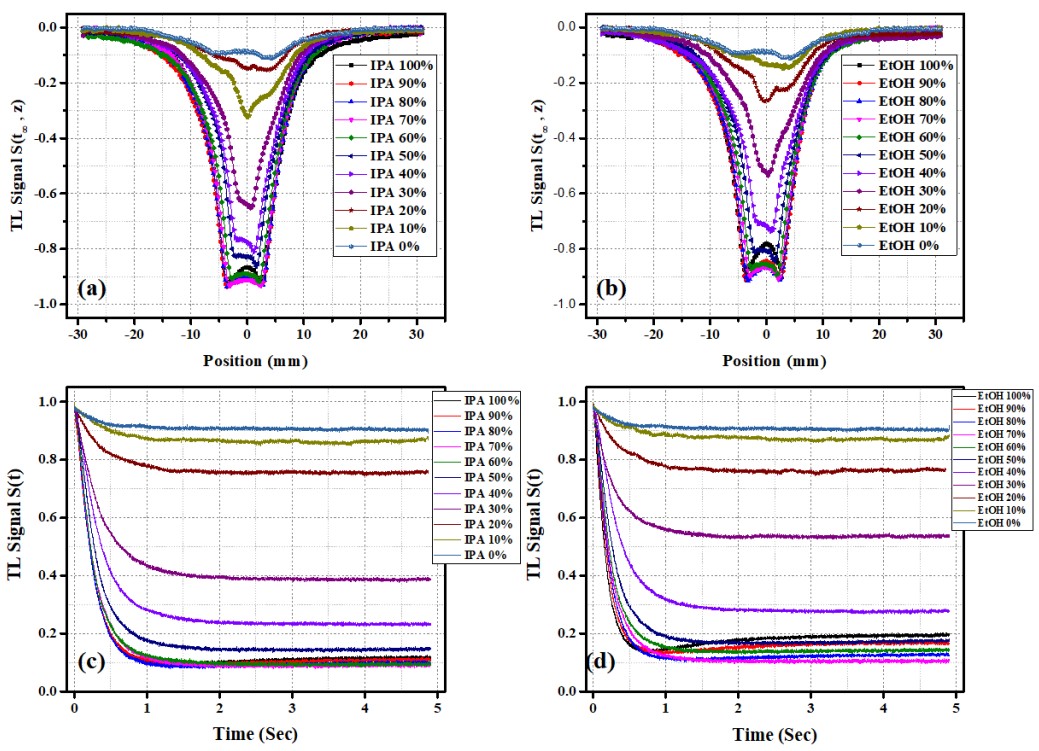

**Figure 6 Dual beam Z-scan.** Dual beam Z-scan traces of (A) IPA-water mixtures, (B) EtOH-water mixtures, time resolved TL signal of (C) IPA-water mixtures, (D) EtOH-water mixtures.

for the pure EtOH solvent. We observe an initial rise in the TL amplitude when water is mixed into the system. Further addition of water in the mixture leads to a decrement in the TL signal.

The time resolved TL signatures of the IPA-water mixture are presented in Fig. 6C. The inflection points in the TL signatures are evidence of the presence of convective mode. When water is added to pure IPA for dilution purposes, we observe that the steady state TL signal of the mixture increases a little compared to the pure IPA. Further addition of water in the mixture eventually leads to a decrement in the steady state TL signal of the mixture. The inflection point and so the convective features are found to be vanishing away slowly with increasing water proportions in the mixture, which could be attributed to the very high thermal conductivity of the water molecules.

Figure 6D represents the time resolved TL signal for EtOH-Water mixtures. The TL response here is also very similar to the earlier one. The steady state TL signal initially increases on water inclusion and then decreases further when the water content exceeds the alcohol content in the mixture. The convective mode of heat transfer is very much present in the mixtures where the alcohol concentration is sufficiently large, but these convective characteristics are diminished at concentrations where water molecules are dominating.

An interesting feature in both systems is observed that the pure solvents do not exhibit the highest steady state TL signal. Rather than the EtOH/IPA (90%)-water (10%) mixture

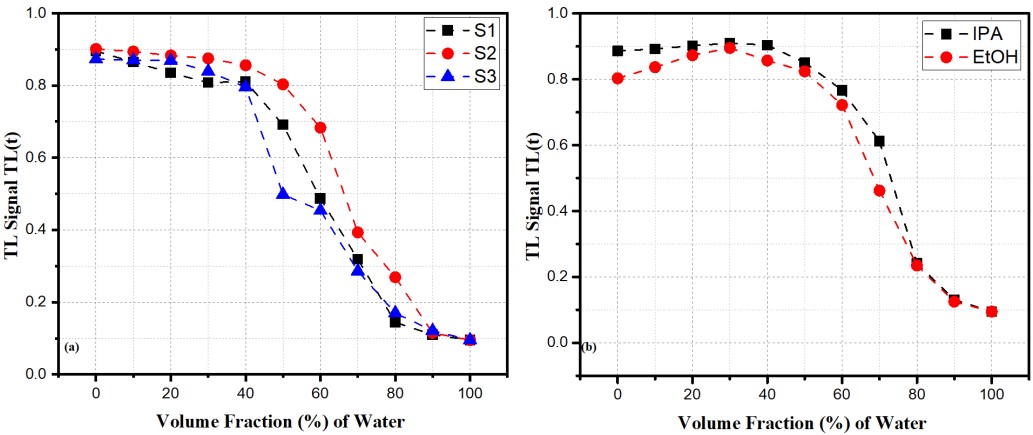

**Figure 7  Variation of state state TL signal.** Variation of the steady state TL signal of (A) sanitizer-water and (B) alcohol-water mixtures at different volumetric ratios.

have a greater TL signal than their pure state. These phenomena arise due to the formation of complex clusters by alcohol and water molecules in the mixtures (*Wakisaka et al., 1998*; *Buhvestov et al., 1998*). These clusters are bigger, have a very complex shape, and are heavier in size than the standalone alcohol molecules. Therefore, the movement and drifting of these structures are very much restricted compared to the small alcohol molecules. As a result, these molecular complexes have minimal contribution to the convective mode of heat transfer, and the accumulated heat is not carried away smoothly as earlier. Hence, we observe a rise in the steady state TL signal at the initial stages of dilution of alcohols with water.

Figure 7 represents the variation of the steady state TL signal of the pure solvents and their diluted mixtures with water. The details on the photothermal response of the sanitizer-water and alcohol-water mixtures using these steady state TL signals are discussed later in the discussion section.

## DISCUSSION

A study of ethanol (EtOH) and isopropyl alcohol (IPA) is necessary to study alcohol-based hand sanitizers because they are the most commonly used active ingredients in these products. A study of EtOH and IPA allows a better understanding of the effect of these ingredients on the accuracy of the measurement technique. This can help to identify any potential interferences that may arise from the presence of these ingredients and to develop appropriate strategies to mitigate them. Additionally, a study of EtOH and IPA also allows a better understanding of these ingredients' effects on the solution's physical properties, such as viscosity, density, and thermal conductivity, which can affect the accuracy of the measurements.

Water dilution significantly affects the effectiveness of hand sanitizers, as it can decrease the concentration of alcohol in the solution and affect the physical properties of the solution. It is important to consider these effects when testing the quality of alcohol-based

hand sanitizers. Studying the alcohol dilution effect with water can provide essential insights into the water dilution effects of sanitizers by directly observing the changes in the concentration of alcohol and its impact on the effectiveness of the hand sanitizer, understanding how the physical properties of the solution are affected by the water dilution. This helps to understand the optimal concentration of alcohol required to kill germs and bacteria effectively and to identify any potential interferences that may arise from the presence of water.

In our study, we have used a simple mixture of water, ethanol and isopropyl alcohol (IPA) to demonstrate the feasibility of our technique. However, commercial hand sanitizers may contain a wide range of other ingredients, such as diols, acrylates, pH buffers, fragrances and thickeners, which may affect the sensitivity of our technique. The potential effects of additives on the sensitivity of our technique will depend on the specific chemical properties of the ingredients and the concentration at which they are used. Some additives may interfere with the thermal measurements and affect the accuracy of our results. For example, certain ingredients such as diols or thickeners may affect the thermal conductivity of the solution, which could lead to inaccurate measurements. Similarly, pH buffers or other ingredients that affect the pH of the solution may affect the accuracy of the measurements. Additionally, some ingredients may have absorption or fluorescence properties that could interfere with the measurement of the thermal lens signal. In case of such interferences, it could lead to a false signal that could be misinterpreted as an alcohol content. It is important to note that these are potential effects, and more research and experiments are required to investigate the actual effect of these ingredients on the accuracy of our technique. It is also important to mention that some additives may not have any significant effect on the sensitivity of our technique, it would depend on the concentration and chemical nature of the ingredient.

Water and ethanol molecules interact through hydrogen bonding. In addition to hydrogen bonding, ethanol and water molecules can also interact through dipole–dipole interactions and London dispersion forces. These interactions are also relatively weak but can contribute to the overall properties of the mixture (*Ghoufi, Artzner & Malfreyt, 2016*; *Quesada Moreno et al., 2020*; *Noskov, Lamoureux & Roux, 2005*).

Hydrogen bonding, dipole–dipole interactions, and London dispersion forces can change the internal structures of ethanol and water molecules and may lead to the formation of clusters. Hydrogen bonding cause to the formation of clusters of water and ethanol molecules. The clusters can be relatively stable and long-lived, and the size and shape of the clusters can depend on the concentration of the molecules and the temperature. The formation of clusters can also change the internal structure of the water and ethanol molecules by altering the distribution of charge within the molecules (*Wakisaka & Matsuura, 2006*; *Mejía, Flórez & Mondragón, 2012*; *Mejía et al., 2007*). The mixture's viscosity and surface tension can be affected by hydrogen-bonded networks formed by water and ethanol molecules. The charge distribution inside water and ethanol molecules is altered by dipole–dipole interactions and London dispersion forces. The heat capacity, thermal conductivity, and convective heat transfer of the mixture are affected by hydrogen bonding and cluster formation in the water and ethanol molecules, which could also
alter photothermal effects. The mixture's thermal conductivity increases with a hydrogen-bonded network upon water inclusion in the mixture (*Putnam et al., 2006*; *Yano, Fukuda & Hashi, 1988*).

At low ethanol concentrations, hydrogen bonding between water molecules is highest, affecting the thermal conductivity and viscosity of a water-ethanol mixture. However, dipole–dipole interactions and London dispersion forces between ethanol molecules dominate the mixture's characteristics at high ethanol concentrations. As ethanol content increases, hydrogen bonding between water molecules decreases. As ethanol concentration rises, ethanol molecules interact more, and there is a nonlinear trend in the viscosity. The viscosity in a water-ethanol mixture initially rises and then decreases due to micro heterogenic effects with increasing mole fraction of ethanol. The maximum viscous region lies within a 0.2–0.4 mole fraction of ethanol in the mixture (*Wakisaka & Matsuura, 2006*; *Khattab et al., 2012*).

Like ethanol, IPA can interact with water through hydrogen bonding and dipole–dipole interactions. In a binary mixture of isopropyl alcohol (IPA) and water, the hydrogen bonding interactions between the hydroxyl group of IPA and the hydroxyl groups of water molecules lead to the formation of clusters of IPA and water molecules. These clusters can affect the heat transfer dynamics of the mixture by acting as barriers to heat transfer, reducing the overall heat transfer coefficient and affecting the thermal conductivity of the mixture. The clusters can be relatively stable and long-lived, and their size and shape can depend on the concentration of the molecules and the temperature (*Guo et al., 2022*). Additionally, the hydrogen bonding can increase the viscosity and surface tension of the mixture, making it more resistant to flow. The heat transfer properties of the mixture are also affected by the concentration of IPA; at low concentrations, the heat transfer properties are dominated by water, but as the concentration of IPA increases, both water and IPA affect the heat transfer properties. The viscosity of the mixture increases with the concentration of IPA, which can result in a decrease in the heat transfer coefficient (*Al-Wahaibi, Grattoni & Muggeridge, 2007*; *Guo et al., 2022*).

In Fig. 7A, we notice the decrement of the steady state TL signal when the volume fraction of water increases. Any amount of water added to the sanitizers changes the photothermal response of the mixture, which is reflected in their TL signature. The decrement of the TL signal is very rapid after a substantial amount of water molecules are present in the system. Figure 7B represents the variation of steady state TL signals of IPA-water and EtOH-Water mixtures at different volumetric proportions of water. For IPA water mixtures, we observe that the steady state TL signal decreases significantly only after the number of water molecules is near the same or larger than the alcohol molecules. The photothermal response of the mixtures is almost like the pure states when the water molecules are in limited numbers in the mixture. We observe a rapid decrease in the steady state TL signal after IPA (50%) — water (50%) composition in the mixture.

For EtOH-water mixtures, we observe that the steady state TL signal increases initially upon the addition of water in the mixture but later, the TL signal is found to be decreasing with increasing water concentration. Pure EtOH doesn't exhibit the highest TL signal due

to its convective features. The initial increase of the TL signal is explained earlier, which is attributed to the formation of complex molecular structures.

The concentration–response curves for the alcohol and water mixture in hand sanitizers exhibit nonlinearity, which is a result of the complex molecular interactions between ethanol, IPA, and water molecules. These interactions lead to the formation of hydrogen-bonded networks and clusters, which alter the internal structure of the molecules and affect their physical and optical properties. Hydrogen bonding between the hydroxyl groups of the ethanol and water molecules causes microheterogeneity in the mixture, leading to the formation of clusters of different sizes and shapes. These clusters act as barriers to heat transfer, reducing the overall heat transfer coefficient of the mixture and affecting the thermal conductivity and viscosity of the mixture. The formation of clusters and hydrogen-bonded networks also leads to changes in the microscopic viscosity of the mixture, which in turn affects the nonlinearity of the concentration–response curves. The nonlinearity of the concentration–response curves is a result of the complex interactions between the molecules in the mixture and the formation of clusters and hydrogen-bonded networks, which are not observable by other methods.

Our TL measurements are sensitive enough to capture any change in the constituents of the mixture. Hence when we dilute the sanitizers with water, the TL measurement is able to reveal the change in its components through a decrement of the steady state TL signal. However, when we compare the dilution effects in sanitizers and in pure solvents, we find that the TL response of the sanitizer mixture is very prompt compared to the other one. In the case of the sanitizers, we do not observe any initial increment of the TL signal. The intermolecular interaction of the main constituent of sanitizer and water is reduced due to the presence of other impurities, like fragrances, glycerin, colorant, *etc.*, in the local environment. Hence the TL response of the sanitizers is different from that of the pure solvents. Consequently, we can identify the change in the composition of the sanitizer mixtures through our TL measurements.

## CONCLUSIONS

In conclusion, our study has presented a novel method for measuring the quality of alcohol based hand sanitizers (ABHS) using femtosecond pulse induced thermal lens spectroscopy. We have shown that by diluting the ABHS with water and performing thermal measurements, it is possible to detect accurately any changes in the alcohol content of the mixture. Our results demonstrate that the thermal lens (TL) signal of the solvent is highly sensitive to changes in alcohol content, making it an effective tool for quality testing of ABHS. The photothermal response was also measured for EtOH and IPA and their mixtures with water to gain greater insight on their intermolecular interactions and dilution effects on heat transfer mechanisms. Our TL measurements appear to be capable of quantifying the available proportions of alcohols in the sanitizer-water mixtures. Our study correlates the effect of ingredients ratio on the performance of the hand sanitizer. This method can be used by manufacturers and regulatory bodies to ensure that ABHS products meet the recommended concentration of alcohol content, and thus maintain

their effectiveness as a hand sanitizer. Overall, our study has highlighted the potential of TL spectroscopy as a useful tool for quality control in the manufacturing and testing of ABHS.

## ACKNOWLEDGEMENTS

We also acknowledge members of FemtoLab for their support during the experiment. All the authors thank Mrs. S Goswami for language correction and editing.

### Funding

This work was supported by the Ministry of Electronics and Information Technology, the Science and Engineering Board of the Department of Science and Technology, and the Space Technology Council of the Indian Space Research Organization of the Govt. of India. The funders had no role in study design, data collection and analysis, decision to publish, or preparation of the manuscript.

### Grant Disclosures

The following grant information was disclosed by the authors:
Ministry of Electronics and Information Technology.
Science and Engineering Board of the Department of Science and Technology.
Space Technology Council of the Indian Space Research Organization of the Govt. of India.

### Competing Interests

Debabrata Goswami is an Academic Editor for PeerJ.

### Author Contributions

- Subhajit Chakraborty performed the experiments, analyzed the data, performed the computation work, prepared figures and/or tables, authored or reviewed drafts of the article, and approved the final draft.
- Ashwini Kumar Rawat performed the experiments, analyzed the data, prepared figures and/or tables, authored or reviewed drafts of the article, and approved the final draft.
- Amit Kumar Mishra performed the experiments, analyzed the data, prepared figures and/or tables, authored or reviewed drafts of the article, and approved the final draft.
- Debabrata Goswami conceived and designed the experiments, authored or reviewed drafts of the article, and approved the final draft.

### Data Availability

  The raw measurements are available in the Supplemental Files.

### Supplemental Information

Supplemental information for this article can be found online at http://dx.doi.org/10.7717/peerj-achem.25#supplemental-information.

![PeerJ]

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
