# Peer review of "Quality assessment of the commercially available alcohol-based hand sanitizers with femtosecond thermal lens spectroscopy"

_PeerJ Analytical Chemistry, doi:10.7717/peerj-achem.25_

## Round 0.1 · original submission · Major Revisions

As you can see, the reviewers point to major flaws in the study. The authors should address the reviewers' comments in full for the manuscript to be considered further.

Reviewer 1 ·

Basic reporting

This is an interesting paper where the authors propose a method to assess the quality of commercially available alcohol-based hand sanitizers using femtosecond thermal lens spectroscopy. The paper lacks data and needs more experiments and a robust hypothesis to make it publishable. Some of the significant points have been included below. This is not an exhaustive list. These comments might help the authors with future submissions in making their research article better scientifically and more impactful.

1) Please proofread the manuscript thoroughly to get rid of any unambiguous and incorrect language.
Examples: Line 164: It should be "whereas" and not "where.": "whereas the main..."
Line 326: PDF: What is PDF? Please introduce abbreviations first before using them.

2)There are some concerns with the references. The manuscript has a total of 39 references that are cited in this manuscript. Out of these, 10 citations are self-citations, which is more than 25% of the total references in this paper, which seems to be a bit excessive. There is an ample amount of recent research by the broad community which could be included, and unnecessary citations could be removed to improve the quality of the paper. There is only one reference from 2022, which is again a self-citation. Multiple recent works relevant to this paper were recently published in 2022, which could significantly improve the quality of the paper if they are researched and linked with these findings.

3) Are femtosecond thermal lens spectroscopy and thermal lens spectroscopy synonymous? If so, please mention it; if not, please highlight the differences. Lines 92-103 do not have sufficient information to help readers understand femtosecond thermal lens spectroscopy. Please add the required information.

4) In Figure 2, the authors can scale the spectra in such a way that it is zoomed to include and highlight the area of interest. For example, the 700 nm to 1050 nm region does not include any information of interest.

Experimental design

5) Methods are not described in sufficient detail. This is a major flaw in this manuscript. The authors do not mention the manufacturer, model, make, product information, etc., of the reagents, instruments, etc, that are used throughout the study. Please include all this information in a uniform format in future submissions, as this is a minimal requirement for the paper to be reproducible by readers if needed. Also, for all the spectroscopic setups used throughout the study, the authors need to provide in detail the instruments used, parts used, and other information in the materials section.

6) In the section beginning in Line 162, S1, S2, and S3, are they commercial? In-house? If in-house, what IPA or Ethanol is used? Manufacturer?

7) Line 175: "we took three sanitizers in their pure state.": What do the authors mean by pure? Please include the "Ingredients" section of the label on these hand sanitizers with the raw data in the future.

8) Why do the authors pick 1560 nm? The ideal choice based on the spectra provided by the authors makes 1500 nm seem like a better one.

Validity of the findings

9) The title of this manuscript is "Quality assessment of the commercially available alcohol-based hand sanitizers with femtosecond thermal lens spectroscopy." The authors, however, have not mentioned in Line 162 if they are using commercial or in-house sanitizers. The authors also need to mention the contents of the hand sanitizers in detail. Commercial hand sanitizers contain various active and inactive ingredients. The authors do not talk about these. These active and inactive ingredients, not ethanol or IPA, might impact the spectroscopic profiles. The authors must perform studies that prove that this method applies to commercial hand sanitizers by performing interference studies and demonstrating the effects of the interfering agents if any.

10) The underlying data that are provided by the authors are very difficult to judge as they are several gaps in the provided information throughout.

11) The conclusion section sounds incomplete. The section only states what experiments the authors performed and contains redundant information. However, the broader implications of the findings of this study, if any, need to be discussed. The conclusion section is not linked to the supporting information provided and discussed in detail.

12) This statement is highly misleading. Line 318: "TL measurements could identify any change in the sanitizer compositions; hence, our technique emerges as a sensitive probe to study such complex systems." The authors have NOT performed their studies in any complex systems in this study. A combination of IPA, Ethanol, and water alone does not mean that these systems are complex. However,m the compositions of commercially available hand sanitizers are complex, but the authors have not paid attention to this complexity in this study. For example, one commercially available hand sanitizer contains these besides IPA and ethanol, Caprylyl Glycol, Glycerin, Isopropyl Myristate, Tocopheryl Acetate, Acrylates/C10-30 Alkyl Acrylate Copolymer, Aminomethyl Propanol, and Fragrance. This is a complex system, and the authors must pay attention to this complexity in their future studies or hypotheses. Please pay attention to the chemistry of the hand sanitizers in your future work and understand the implications of that in your study.

Additional comments

13) Unfortunately, the article in its present state fails in all the above three sections even though the work's title and idea are interesting. If the authors can address these comments, provide requested information, perform numerous critical experiments, explain this from an analytical chemistry perspective and provide comparisons that are lacking and re-submit this in the future, this manuscript may be considered for review again. Since there are several fundamental flaws, I would not recommend this article for publication in its present form.

Reviewer 2 ·

Basic reporting

The work presents application of a novel spectroscopic technique to a recent global health issue-related analytical need. However, recent work addressing the research problem of analyzing hand sanitizers is omitted in the manuscript:
1. Abrigo, Nicolas, et al. "Development and validation of a headspace GC-MS method to evaluate the interconversion of impurities and the product quality of liquid hand sanitizers." AAPS open 8.1 (2022): 1-13.
2. Abrigo, Nicolas, et al. "Application of a headspace GC–MS method to evaluate the product quality of alcohol‐based hand wipe sanitizers." Biomedical Chromatography 36.10 (2022): e5432.
3. Gupta, Nirzari, Jason D. Rodriguez, and Huzeyfe Yilmaz. "Through-container quantitative analysis of hand sanitizers using spatially offset Raman spectroscopy." Communications Chemistry 4.1 (2021): 1-9.

The manuscript also has numerous grammatical errors. It is partly written in non-professional English with narration switches within certain paragraphs.

Experimental design

While the idea of utilizing thermal lens spectroscopy for characterization of alcohol-based hand sanitizers is novel, the results are highly underwhelming. Authors frame the research question with limitations of current techniques in tedious sample preparation, expensive instrumentation and sensitivity yet the experimental design and the results address neither of the above three issues any better than the most recent approaches (references listed above).

An additional issue is with the sample preparation. It is not clear what the hand sanitizers S1-S3 are. Did the authors purchase commercial hand sanitizer samples? Do the authors know the quantitative make up of these hand sanitizer formulations?

a. Despite generating concentration-response curves for both the samples and IPA-Ethanol mixtures authors do not attempt to assess the quantification capability of their technique. A simple regression model may allow readers to understand how well thermal lens spectroscopy performs against GC, MS, NIR, Raman, etc.
b. Concentration-response curves are highly nonlinear however reader is unable to find any explanation for this effect. The discussion of the results should be re-considered for the entirety of the manuscript.
c. Figure 6d is missing in the main manuscript file.
d. Not all z-scan and time-resolved measurements need to be demonstrated for every measurement. Figures are generally cluttered with the same information presented twice. Z-scan measurements may be moved to supplementary information.

Validity of the findings

The work defines a research problem for quantification of alcohol amount in hand sanitizers based on the limitations of the current techniques. While the experimental work is robust and sound, the lack of any quantitative analysis from thermal lens spectroscopy is an important flaw. Since the authors initially fail to review the literature appropriately they may need to reassess the benefit of their work if any other than using a novel technique. For instance, highly sensitive GC-MS methods have been presented recently (above references) for impurity analysis or highly practical through container SORS method have been shown to quantify alcohol amounts in hand sanitizers. Present work can be considered neither practical given the experimental design, nor sensitive given the concentration-response curves.
Overall, this work requires additional experiments with more samples and more detailed sample preparation (such as origin information, ingredients, etc.), more detailed discussion of the results with more succinct figures (eliminating redundant data), at the minimum a regression analysis of some sort to predict alcohol concentrations (a calibration dataset may be required), and correcting the writing and grammatical errors throughout the manuscript.

Reviewer 3 ·

Basic reporting

1. Clear and unambiguous, professional English used throughout.

Yes.
Line 164 -165: S1 is an IPA-based sanitizer where the main constituent of S2 is ethyl alcohol (EtOH) --> Should be whereas
Line 178 - 't' is missing in 'to'
Please read through carefully and do a spell and grammar check to avoid unintended capital letters in-between sentences

2. Literature references, sufficient field background/context provided.

Yes sufficient literature on TL techniques is provided. However, I would recommend adding a few more sentences if available on TL technique used in alcohol based systems in literature.

3. Professional article structure, figures, tables. Raw data shared.
Yes

4. Self-contained with relevant results to hypotheses.
Yes

Experimental design

1) Original primary research within Aims and Scope of the journal.

Yes

I would recommend bumping the lines 175-180 to sample preparation for ease of understanding.

2) Research question well defined, relevant & meaningful. It is stated how research fills an identified knowledge gap.

Yes

3) Rigorous investigation performed to a high technical & ethical standard.
Yes

4) Methods described with sufficient detail & information to replicate.
Yes

Validity of the findings

1) All underlying data have been provided; they are robust, statistically sound, & controlled.

Yes.

2) Impact and novelty not assessed. Meaningful replication encouraged where rationale & benefit to literature is clearly stated.

Yes.
3) Conclusions are well stated, linked to original research question & limited to supporting results.

Yes

Additional comments

Overall a very good manuscript and well- defined DOE. Kudos to the authors for a well laid out manuscript.
My recommendations are:

1) Could you please split up the results and discussion part into subheadings so it is easy on the reader. In its present form its very lengthy and reader might end up losing interest along the way.

2) What would happen to the sensitivity of the measurements if additives are present in the sanitizer. Typical store bought formulations contain diols, acrylates, pH buffers, fragrances and thickeners - how would these ingredients affect the sensitivity of the techniques? Clarification on these aspects would help in better understanding.

---

## Round 0.2 · Minor Revisions

Please consider the comment of Reviewer 3.

Reviewer 2 ·

Basic reporting

Revision addressed the previous concerns and issues. Current version is written well, has all the relevant references and background.

Experimental design

Research questions reframed to fit the results and where this work may have value.

Validity of the findings

Benefit to literature may be marginal however positive. Conclusions are brought to the level of support from the results. Data is robust and sound.

Reviewer 3 ·

Basic reporting

Meets expectation

Experimental design

Meets Expectation

Validity of the findings

Meets Expectation

Additional comments

My only comment - would be incorporating the below information in the manuscript either in discussion or introduction, so readers are aware of the limitations as well

2) What would happen to the sensitivity of the measurements if additives are present in the sanitizer. Typical store bought formulations contain diols, acrylates, pH buffers, fragrances and thickeners - how would these ingredients affect the sensitivity of the techniques? Clarification on these aspects would help in better understanding.

Answer
In our study, we have used a simple mixture of water, ethanol and isopropyl alcohol (IPA) to demonstrate the feasibility of our technique. However, commercial hand sanitizers may contain a wide range of other ingredients, such as diols, acrylates, pH buffers, fragrances and thickeners, which may affect the sensitivity of our technique.
The potential effects of additives on the sensitivity of our technique will depend on the specific chemical properties of the ingredients and the concentration at which they are used. Some additives may interfere with the thermal measurements and affect the accuracy of our results.

For example, certain ingredients such as diols or thickeners may affect the thermal conductivity of the solution, which could lead to inaccurate measurements. Similarly, pH buffers or other ingredients that affect the pH of the solution may affect the accuracy of the measurements.
Additionally, some ingredients may have absorption or fluorescence properties that could interfere with the measurement of the thermal lens signal. In case of such interferences, it could lead to a false signal that could be misinterpreted as an alcohol content.
It is important to note that these are potential effects, and more research and experiments are required to investigate the actual effect of these ingredients on the accuracy of our technique. It is also important to mention that some additives may not have any significant effect on the sensitivity of our technique, it would depend on the concentration and chemical nature of the ingredient.

---

## Round 0.3 · accepted · Accept

The reviewer has recommended acceptance.

Reviewer 3 ·

Basic reporting

n/a

Experimental design

n/a

Validity of the findings

n/a

Additional comments

n/a